# Anomaly Detection in Airborne Fourier Transform Thermal Infrared Spectrometer Images Based on Emissivity and a Segmented Low-Rank Prior

**Xuhe Zhu** [1,2], **Liqin Cao** [2,3,*], **Shaoyu Wang** [1,2], **Lyuzhou Gao** [1,2] and **Yanfei Zhong** [1,2]

1   State Key Laboratory of Information Engineering in Surveying, Mapping and Remote Sensing, Wuhan University, Wuhan 430079, China; zhuxuhe@whu.edu.cn (X.Z.); wangshaoyu@whu.edu.cn (S.W.); gaolyuzhou@whu.edu.cn (L.G.); zhongyanfei@whu.edu.cn (Y.Z.)
2   Hubei Provincial Engineering Research Center of Natural Resources Remote Sensing Monitoring, Wuhan University, Wuhan 430079, China
3   School of Printing and Packaging, Wuhan University, Wuhan 430079, China
*   Correspondence: clq@whu.edu.cn; Tel.: +86-27-68778529

**Abstract:** Although hyperspectral anomaly detection is commonly conducted in the visible, near-infrared, and shortwave infrared spectral regions, there has been less research on hyperspectral anomaly detection in the longwave infrared (LWIR) hyperspectral region. The radiance of thermal infrared hyperspectral imagery is determined by the temperature and emissivity. To avoid the detection uncertainty caused by the single factor of temperature, emissivity can be introduced to detect anomalies. However, in the emissivity domain, the spectral contrast and signal-to-noise ratio (SNR) are low, which makes it difficult to separate the anomalies from the background. In this paper, an anomaly detection method combining emissivity and a segmented low-rank prior (EaSLRP) is proposed for use with thermal infrared hyperspectral imagery. The EaSLRP method is divided into three parts—(1) temperature/emissivity retrieval, (2) extraction of the thermal infrared hyperspectral background information, and (3) Mahalanobis distance detection. A homogeneous region generation method is also proposed to solve the problem of the complex global background leading to inaccurate background estimation. The GoDec method is used for matrix decomposition and background information extraction and to remove some of the noise. The proposed Mahalanobis distance detector then uses the background component and original image for anomaly detection, while highlighting the spectral difference between the anomalies and background. This method can also suppress the influence of noise, to some extent. The experimental results obtained with airborne Fourier transform thermal infrared spectrometer hyperspectral images demonstrate that the EaSLRP method is effective when compared with the Reed–Xiaoli detector (RXD), the segmented RX detector (SegRX), the low-rank and sparse representation-based detector (LRASR), the low-rank and sparse matrix decomposition (LRaSMD)-based Mahalanobis distance method (LSMAD), and the locally enhanced low-rank prior method (LELRP-AD).

**Keywords:** hyperspectral longwave infrared (LWIR) imagery; low-rank prior; imagery segmentation; anomaly detection

## 1. Introduction

Hyperspectral remote sensing target detection has developed rapidly in recent years and has become an effective means to remotely detect targets of interest [1–3]. Hyperspectral remote sensing imagery contains rich spectral information, and the high-dimensional features can effectively support the identification of different targets. Hyperspectral anomaly detection has broad prospects for civil and military applications, such as mineral exploration, border monitoring, search and rescue, military reconnaissance, etc. Many hyperspectral target detection methods have been proposed for use in the visible and

shortwave infrared regions [4]. However, there has been less research on hyperspectral target detection in the longwave infrared (LWIR) hyperspectral region. In recent years, the scientific community has begun to pay attention to the application of the LWIR hyperspectral region and the problems of target detection, such as military reconnaissance at night. Differing from the visible and shortwave infrared regions, the radiance in the LWIR spectral region, as measured by the sensor, mainly comes from the material itself and the atmosphere. According to the radiative transfer equation, the radiance is determined by the temperature and emissivity. The thermal infrared systems allow day and night operation and can provide us with the surface temperature information of objects [5]. Both the temperature and emissivity information can be used to assist with target detection.

Hyperspectral target detection can be divided into hyperspectral anomaly detection and target-signature-based target detection [6]. In the LWIR region, the research has been more focused on signature-based target detection. The first step in target detection is an atmospheric correction. Moderate resolution atmospheric transmission (MODTRAN) [7] or the in-scene atmospheric correction (ISAC) algorithm [8] is applied to the original radiance imagery to estimate the upward radiance and spectral transmittance, and then a temperature-emissivity separation (TES) algorithm such as ASTER's TES algorithm [9] or the maximum smoothness TES algorithm [10] is applied to remove the downward radiance and estimate the surface temperature and spectral emissivity. The next step is to use a statistical algorithm to detect the target of interest in the emissivity domain or radiance domain according to the signature of the target. Statistical algorithms such as the spectral matched filter (SMF) [11], spectral angle mapper (SAM) [12], generalized likelihood ratio test (GLRT) [13], and adaptive coherence estimator (ACE) [14] have been used for detection in the LWIR region. However, previous studies have shown that the spectral emissivity change in the LWIR region is usually less than the corresponding spectral reflectance change in the reflectance region [15]. Therefore, some researchers have added a material identification algorithm to the target detection algorithm to achieve hybrid spectral analysis [16]. Although these signature-based target detection methods have achieved good results in LWIR region target detection, there are still some problems to be addressed. On the one hand, due to the particularity of the LWIR region, the measurement equipment is particularly sensitive to the surrounding environment, which can result in large measurement errors. On the other hand, LWIR region target detection is limited by the accuracy of the atmospheric compensation (AC) algorithm and TES algorithm, and the obtained temperature and emissivity may not be accurate, which makes it difficult for signature-based target detection and identification algorithms to match the measured spectrum with the image spectrum.

Hyperspectral anomaly detection does not require prior knowledge of the target and background, and only considers the difference of the spectral characteristics between pixels. Many anomaly detection algorithms have been proposed for use in the visible and near-infrared (VNIR) to shortwave infrared (SWIR) region, and these algorithms can also be used in the LWIR region. Statistical methods such as the Reed–Xiaoli detector (RXD) [11] assume that the image background is a multi-dimensional Gaussian random process with fast spatial mean change and slow variance change, which is a suitable assumption for anomaly target detection when the background distribution is relatively simple. However, since the background statistical variables are calculated based on the whole image, they are inevitably affected by the anomaly targets. Inspired by the RXD method, many improved methods have now been proposed. For example, to solve the problem of anomaly contamination caused by global computing, the segmented RX detector (SegRX) [17] first divides the image into blocks and then applies the RX detector in these blocks. However, the statistical methods have a common problem in that the statistical distribution cannot accurately describe the real background [18].

To avoid the inaccurate statistical distribution of the background, representation-based methods have been proposed, including hyperspectral anomaly methods based on collaborative representation (CR) [19] and sparse representation (SR) [20]. These representation-

based methods have achieved good results. However, the background calculation involves the anomaly pixels, which still affect the detection effect to a certain extent. More recently, researchers have begun to apply the low-rank prior to hyperspectral anomaly detection [21–24]. This approach can extract the background knowledge and anomalous knowledge from the hyperspectral imagery at the same time. The low-rank-based methods assume that the background pixel vector can be approximately represented as a linear combination of several groups of basis vectors in the low-dimensional subspace, while the anomalous pixel vector cannot, hence the background has low-rank characteristics. Compared with the background pixels, the proportion occupied by anomalies in hyperspectral imagery is very small.

In LWIR hyperspectral imagery, the spectral contrast and SNR are low [15]. Weak features are easily disturbed by noise, which makes it difficult to separate the target from the background. If the detection algorithm is directly applied to the global image, the background information estimation will be inaccurate due to the interference of abnormal pixels and image noise. As a result, the spectral contrast between the background and anomalies will be further reduced, which makes the anomalies and background harder to separate.

In this paper, in order to better distinguish the anomalies and background in airborne Fourier transform thermal infrared spectrometer hyperspectral images, an anomaly detection method combining emissivity and a segmented low-rank prior (EaSLRP) is proposed. We propose an anomaly detection method for the emissivity domain of LWIR hyperspectral imagery and address the problems of the low spectral contrast and SNR. Furthermore, we describe how we used an advanced airborne LWIR hyperspectral instrument to provide the experimental data used to verify the proposed method.

The main contributions of this paper can be summarized as follows:

(1) The proposed low-rank-based method is divided into three parts, which are (1) temperature/emissivity retrieval, (2) extraction of the thermal infrared hyperspectral background information, and (3) Mahalanobis distance detection. The proposed method can better represent the complex background of thermal infrared hyperspectral imagery than the methods based on statistical distribution assumptions, and it solves the problems of the low spectral contrast and SNR;

(2) In the process of extracting the thermal infrared hyperspectral background information, the Potts method [25] is introduced to segment the radiation image, and the temperature information is combined to further determine the boundary of the segmentation area. The segmentation boundary is adopted for the emissivity image to achieve regional segmentation, and then the local emissivity data is decomposed by the GoDec method [26] to remove some noise and obtain more accurate background information;

(3) In the part of Mahalanobis distance detection, the background component and the original data are utilized by the Mahalanobis distance detector to detect the anomalies in the LWIR hyperspectral imagery. The proposed EaSLRP method can thus highlight the spectral differences between the anomalies and background.

The rest of this paper is organized as follows. In Section 2 the low-rank and sparse model for the LWIR region is presented, the theory of LWIR hyperspectral anomaly detection and the proposed method are described. In Section 3 the experimental data are introduced and the experimental results are analyzed. In Section 4 the data and parameters are analyzed. Finally, our conclusions are summarized in Section 5.

## 2. Method

### 2.1. Low-Rank Model

In thermal infrared hyperspectral imagery, the background occupies most of the image area and is both continuous and smooth. In contrast to the background, the anomalies

occupy a very small proportion of the hyperspectral imagery, and possibly only a few pixels. For a pixel $x_i$ in the $k$th band, it can be modeled as

$$x_i = b_i + s_i \tag{1}$$

where $b_i$ represents the background and $s_i$ represents the anomaly. If $x_i$ is an anomaly, then the background component $b_i$ equals zero; otherwise, the anomaly component $s_i$ equals zero. In the $k$th band of the image, most of the area is the background, and the anomalies are sparse. The $k$th band of the image is also composed of anomalies and background.

For thermal infrared hyperspectral imagery, the spectral vector $X_i = [x_i^1, \ldots, x_i^n]$ consists of the $x$ in each band, where $n$ is the number of bands. $X_i$ then represents the composition of the background and anomaly part of each band. This can be shown as

$$X_i = B_i + S_i \tag{2}$$

where $B_i = [b_i^1, \ldots, b_i^n]$ and $S_i = [s_i^1, \ldots, s_i^n]$. $S_i$ represents the anomaly feature of the pixel vector $X_i$. However, the elements in $S_i$ might be zero.

For real thermal infrared hyperspectral imagery, the image data $X = [X_I, \ldots, X_N]^T \in R^{N \times M}$ can be represented as the combination of the background component $B = [B_I, \ldots, B_N]^T$ and anomaly component $S = [S_I, \ldots, S_N]^T$, as shown in (3),

$$X = B + S \tag{3}$$

where $N$ and $M$ are the numbers of pixels and bands of the image, respectively. The background component $B$ contains the background information of the LWIR hyperspectral imagery, and the anomaly component $S$ contains the anomaly information. However, real hyperspectral data are inevitably affected by noise $N$. Real data can thus be denoted as

$$X = B + S + N \tag{4}$$

Previous research has shown that hyperspectral data have low intrinsic dimensionality [27]. In other words, most of the pixel vectors in hyperspectral images can be represented by several basis vectors, and only a few of the pixel vectors cannot be represented in this way. The pixel vectors that cannot be represented by several basis vectors are mainly affected by gross sparse errors or outliers. In a homogeneous region, the spectra of the background pixels are highly similar, so that a background pixel can be approximately represented by the surrounding pixels. Therefore, the background has low-rank characteristics. As the proportion of anomaly pixels is small, they are sparse in this homogeneous region. According to the above hypothesis, the real hyperspectral data $X$ can be represented as

$$X = L + S + N \tag{5}$$

where $L$ is the low-rank background part, $S$ is the sparse anomaly part, and $N$ is the noise part. The low-rank background part $L$ represents the global background information, and the sparse anomaly part represents the global anomaly information.

### 2.2. LWIR Hyperspectral Anomaly Detection

Although the low-rank model can describe the background distribution of the thermal infrared hyperspectral imagery, the following problems still exist when processing the global image with the target detection algorithm. Firstly, the complex background makes the description of the background inaccurate, which affects the detection accuracy. However, detection in the local area after blocking leads to an increase in the proportion of anomaly pixels and a decrease in the proportion of background pixels, which affects the accuracy of low-rank and sparse decomposition. Meanwhile, due to the low spectral contrast and SNR of thermal infrared imagery, the effect of an anomaly detection method based on the sparse components is not stable, and it cannot distinguish anomalies and

background well. Therefore, a local low-rank prior anomaly detection is proposed in this paper to address these problems.

To overcome the problems caused by the complex background of the global image, the original image is divided into several homogeneous regions. Because there is no prior knowledge of the background and anomalies, the Potts-based method is introduced as the region segmentation method. Generating multiple homogeneous regions can reduce the background complexity of each region. A background component that is calculated in homogeneous regions is simpler. As the number of pixels in the homogeneous region decreases and the anomaly pixels in the homogeneous region does not change, the proportion of anomaly pixels increases. Hence, the sparsity of the abnormal pixels decreases, which affects the subsequent detection results.

It is therefore necessary to enhance the sparsity of the anomalies in homogeneous regions. In the proposed approach, a new matrix is built to replace the original local data matrix for low-rank background information extraction.

The low-rank background part is selected to estimate the background covariance matrix. The Mahalanobis distance is then introduced to detect the anomalies in the original local image.

The flowchart of the proposed EaSLRP method is shown in Figure 1.

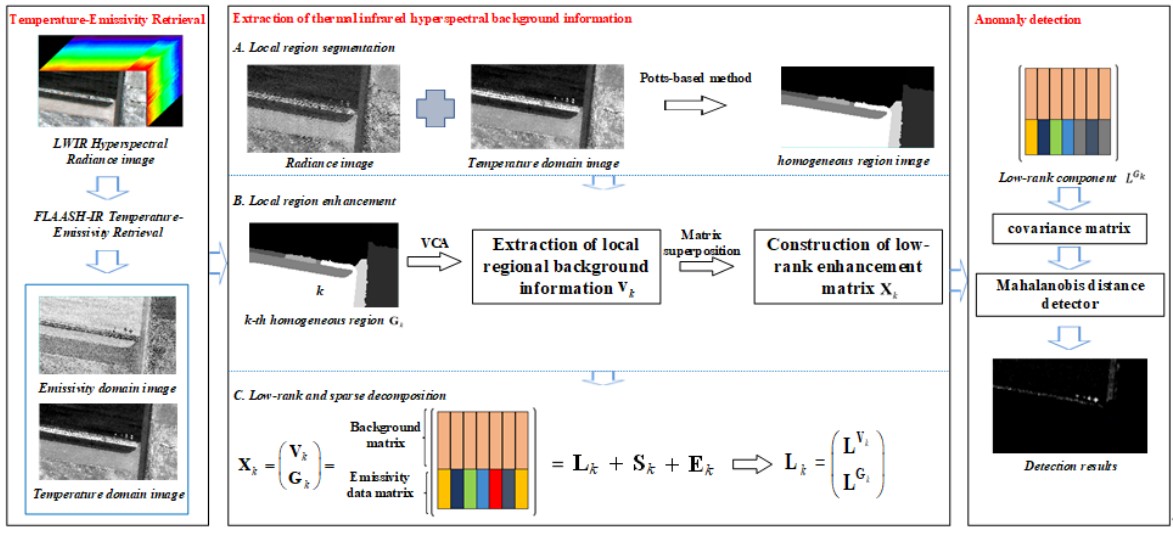

**Figure 1.** Flowchart of the proposed emissivity and a segmented low-rank prior (EaSLRP) method.

### 2.2.1. Thermal Infrared Hyperspectral Radiance Signal Model and Temperature/Emissivity Retrieval

The spectral emissivity can be used to characterize different materials, and the spectral feature can be used to detect anomalies.

According to the radiative transfer model (RTM), the radiation received by an LWIR hyperspectral sensor can be represented a

$$L_{sen} = [\varepsilon(\lambda)B(\lambda, T) + (1 - \varepsilon(\lambda))L_d(\lambda)]\tau(\lambda) + L_u(\lambda) \tag{6}$$

where $L_{sen}$ is the radiance measured by the sensor; $\varepsilon(\lambda)$ is the emissivity; $B(\lambda, T)$ is the radiance of a black body at temperature $T$, which is obtained from the Planck function; $L_d(\lambda)$ is the downwelling atmospheric radiance; $\tau(\lambda)$ is the atmospheric transmission; and $L_u(\lambda)$ is the atmospheric upwelling radiance.

Due to the particularity of the LWIR region, thermal infrared hyperspectral equipment is easily disturbed by the atmosphere and other factors. In order to accurately separate the background and anomalies, atmospheric correction must be carried out for the radiance to remove the influence of the atmosphere. MODTRAN is used to estimate the atmospheric upwelling radiance and atmospheric transmission. The TES algorithm is then used to

remove the downwelling atmospheric radiance, and the temperature and surface emissivity are then obtained. The temperature image is a single band image, and the emissivity image has the same number of bands as the original radiance image. In this study, the fast line-of-sight atmospheric analysis of spectral hypercubes–infrared (FLAASH–IR) algorithm [28] was chosen to retrieve the temperature and the emissivity.

The following algorithms are applied in the emissivity domain.

### 2.2.2. Extraction of Thermal Infrared Hyperspectral Background Information

For thermal infrared hyperspectral imagery, the background of a homogeneous region is more unitary than the original global image. To simplify the background of the imagery and estimate the anomalies accurately from the homogeneous background information, the thermal infrared hyperspectral imagery is segmented based on the Potts-based method.

In the radiance domain of the thermal infrared imagery, the gray value of the image changes very sharply in the boundary regions. For the temperature domain image in particular, although the temperature will be similar in a homogeneous region, the temperature can be significantly different in the boundary areas because of the change of the cover types. We can therefore use this feature to segment the region. However, this change is not obvious in the emissivity domain. Therefore, the radiance domain of the thermal infrared hyperspectral imagery is selected for the region segmentation, and the temperature image is used to enhance the boundary difference. The Potts-based method's minimization problem can be expressed as

$$u^* = \underset{u}{\mathrm{argmin}}\,\gamma \cdot \|\nabla u\|_0 + \|u - f_{\mathrm{PCA}}\|_2^2 \tag{7}$$

Because the region segmentation boundaries are compatible with the support of the gradient $\nabla u$, $\|\nabla u\|_0$ represents the boundary length of the segmented region and is called the boundary term. $f_{\mathrm{PCA}} \in \mathbb{R}^{3 \times N}$ is the superposition of the first two bands obtained by principal component analysis of the original radiance image, and one band is the temperature domain image. Figure 2 shows the superimposed images.

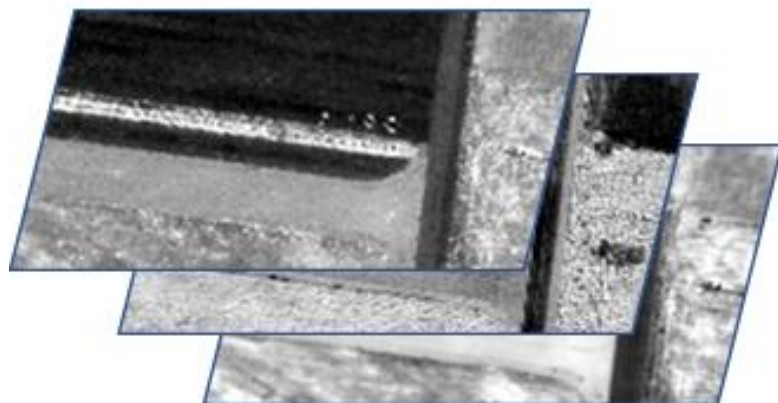

**Figure 2.** Superimposed images.

$u$ is the piecewise constant function, whose dimension is the same as $f_{\mathrm{PCA}}$. $\gamma$ is the empirical model parameter. Figure 3 shows the infrared hyperspectral image and the segmentation results. Note that, if the temperature information is not used, the radiance image can still complete the region segmentation, but some regions may not be segmented. The temperature information can strengthen the boundary information of different regions. The image is then divided into $k$ homogeneous regions.

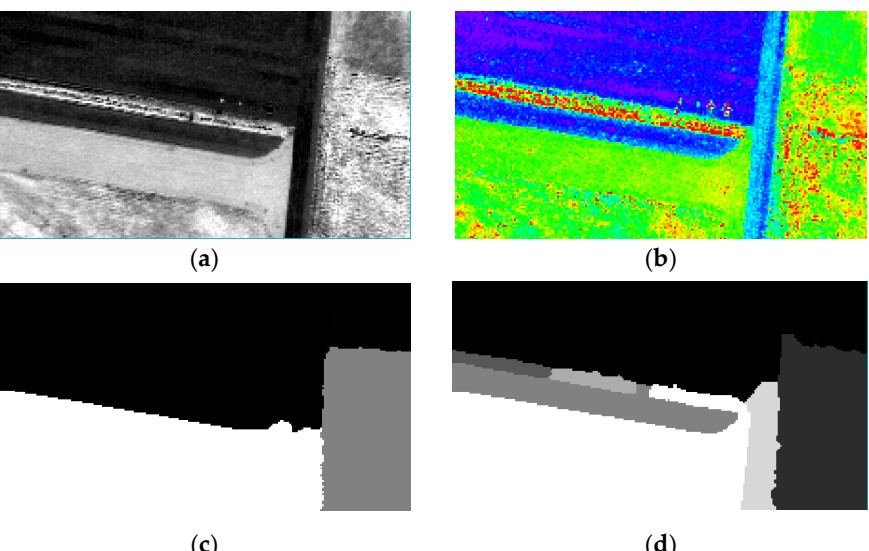

**Figure 3.** The infrared hyperspectral image and the segmentation results. (**a**) Longwave infrared (LWIR) hyperspectral image; (**b**) temperature image; (**c**) segmentation without temperature; and (**d**) segmentation with temperature.

The radiance domain signatures of thermal infrared hyperspectral images are determined by the material temperature and emissivity. To remove the influence of inaccurate temperature estimation and better separate the background and anomalies, the emissivity domain image is selected to carry out the subsequent operations.

After the completion of homogeneous region segmentation, the proportion of anomaly pixels in the local region increases, which affects the results of the low-rank background information extraction. Therefore, it is necessary to enhance the sparsity of the anomaly pixels in the local region to improve the decomposition result. In the proposed approach, a new matrix is constructed to replace the original local emissivity data matrix for sparse decomposition and address the problem of the local data matrix. The new matrix is composed of the local emissivity data matrix and the background endmember matrix of the corresponding region. The background endmember matrix is composed of the spectral vectors of the background endmembers, and the background endmember spectral vectors are obtained by endmember extraction in the local region. Each column of the background endmember matrix is a column vector composed of the background endmember spectra of the corresponding region. Therefore, the rank of the background endmember matrix is one, which keeps the low-rank principle of the designed matrix. Because only the background endmember spectra are utilized, the additional matrix is independent of the anomalies, which makes the decomposition more accurate.

Because the background information extraction strategy is the same in each homogeneous region, the $k$th homogeneous region is taken as an example for a better explanation. According to our idea, the newly designed matrix can be expressed as

$$\mathbf{X}_k = \left( \begin{array}{c} \mathbf{V}_k \\ \mathbf{G}_k \end{array} \right) \in \mathbb{R}^{((r+1) \times B) \times N_k} \tag{8}$$

where $\mathbf{X}_k$ is the designed matrix of the $k$th homogeneous region. $\mathbf{V}_k$ is a background endmember matrix of rank 1, which is obtained by an endmember extraction method, e.g., vertex component analysis (VCA). $\mathbf{X}_k$ is as shown in Figure 4.

For the newly designed matrix, only the number of low-rank background elements is increased, and the number of anomaly elements is unchanged. Therefore, compared with the original local region, the new matrix can better highlight the sparsity of the anomaly pixels, which makes the subsequent low-rank background information extraction more accurate.

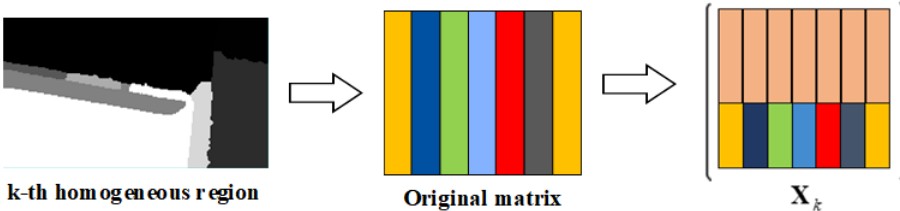

**Figure 4.** Illustration of low-rank and sparse decomposition.

After the construction matrix is designed, the construction matrix is used to decompose the low-rank background part. In order to separate the low-rank components, sparse components, and noise, the GoDec algorithm is adopted for the decomposition. The GoDec algorithm utilizes faster bilateral random projections (BRPs) in randomized approximate matrix decomposition (RAMD). The process of low-rank background information extraction is shown in Figure 5.

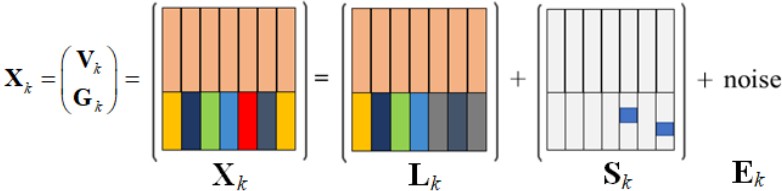

**Figure 5.** Illustration of low-rank and sparse decomposition. Where $\mathbf{L}_k$ is the low-rank part, $\mathbf{S}_k$ is the sparse anomaly part, and $\mathbf{E}_k$ is the noise part.

Since the image has been segmented into multiple homogeneous regions with a simpler background, and the sparsity of the anomalies of the data matrix of each homogeneous region has been enhanced, the image background information is obtained accurately after low-rank and sparse decomposition. Meanwhile, the noise is removed from the enhanced matrix, and the low-rank background information will be less affected by noise.

2.2.3. Mahalanobis Distance Detection

After the low-rank background information extraction, the background component and sparse component are obtained.

$$\mathbf{L}_k = \begin{pmatrix} \mathbf{L}^{\mathbf{V}_k} \\ \mathbf{L}^{\mathbf{G}_k} \end{pmatrix} \in \mathbb{R}^{((r+1)\times B)\times N_k} \tag{9}$$

where $\mathbf{L}_k$ is the low-rank background matrix, which consists of two parts $\mathbf{V}_k$ and $\mathbf{G}_k$. To avoid the interference of redundant information, the $\mathbf{G}_k \in \mathbb{R}^{B \times N_k}$ part is selected to estimate the background covariance matrix of the local homogeneous region. Therefore, the anomaly detector based on the Mahalanobis distance can be written as

$$D(x) = (x_k - \mu_k)^T \Gamma_k^{-1}(x_k - \mu_k) \tag{10}$$

where $x_k$ is the pixel vector of the original emissivity image, $\mu_k$ is the mean emissivity vector of the background data, and $\Gamma_k$ is the covariance matrix of the background data. The background component $\mathbf{L}^{\mathbf{G}_k} = [\mathbf{L}_1^{\mathbf{G}_k}, \mathbf{L}_2^{\mathbf{G}_k}, \ldots, \mathbf{L}_N^{\mathbf{G}_k}]$, and N is the number of pixels in the homogeneous region. $\mu_k$ and $\Gamma_k$ can then be written as

$$\mu_k = \frac{1}{N}(\mathbf{L}_1^{\mathbf{G}_k} + \mathbf{L}_2^{\mathbf{G}_k} + \cdots + \mathbf{L}_N^{\mathbf{G}_k}) \tag{11}$$

$$\Gamma_k = \frac{1}{N}(\mathbf{L}^{\mathbf{G}_k} - \mu_k)^T(\mathbf{L}^{\mathbf{G}_k} - \mu_k) \tag{12}$$

Many anomaly detection methods have verified the validity of the Mahalanobis distance detector, including the RXD method. In the proposed method, the Mahalanobis distance detector is adopted, which extracts the background information of the thermal infrared imagery. The purer background information can thus enhance the difference between the anomalies and background.

## 3. Results

### 3.1. Experimental Measurements

The experimental area is in Shangjie, Zhengzhou, Henan Province, China, and is about 14 km$^2$ in area. The data were collected from 2.5 km above the ground using the state-of-the-art thermal infrared imaging spectrometer called the Hyper-Cam-LW, which is made by Telops. The data were collected on 30 March 2019. The Telops Hyper-Cam-LW was installed on a stable platform. The GPS and inertial measurement unit (IMU) were also installed on the platform and used for geographic referencing and tracking when flying. During the data acquisition, the image motion compensation mirror used the GPS/IMU data to effectively compensate for the aircraft motion.

The ground-based Hyper-Cam-LW is a Fourier transform imaging spectrometer. Its spectral resolution can reach 0.25 cm$^{-1}$, and its average noise equivalent spectral radiance at 10 μm is 24.2 nW/cm$^2 \cdot$ sr $\cdot$ cm$^{-1}$. It uses a $320 \times 256$ long-wavelength infrared mercury cadmium telluride (MCT) photovoltaic (PV) focal plane array detector, which can be windowed and formatted to fit the required size and reduce the acquisition time. The LWIR spectral range of the obtained imagery is 8–12.5 μm, and the imagery has 81 bands. The spatial resolution of the thermal infrared imagery was 0.95 m. All the image scene sizes were $227 \times 125$ pixels. In order to reduce the atmospheric interference as much as possible, we eliminated some bands with serious atmospheric interference and selected 78 bands for the research.

To verify the effectiveness of the EaSLRP algorithm in detecting anomalies, three sets of LWIR hyperspectral data were selected from the Shangjie data sets. The Ground-truth maps are based on field investigation and labeled with recorded longitude and latitude information combined with the aerial flight images.

### 3.2. Experimental Methods and Parameter Settings

Both classical algorithms and newly developed algorithms were chosen as the comparison algorithms. The statistical anomaly detection methods were the RXD method [11] and the SegRX method [17]. The anomaly detection methods based on the low-rank principle were the LELRP-AD(LSMAD) [21] and the locally enhanced low-rank prior (LELRP-AD) method [24]. The low-rank and sparse representation (LRASR) method [23] was selected as a method based on low-rank representation.

The RXD method does not require us to set any parameters. In order to make the results more convincing, SegRX used the same segmentation regions as EaSLRP.

For LSMAD, we set the cardinality $k$ for all three data sets to 0.004. In the method description of LSMAD, the value of rank $r$ is generally between 1 and 2. Considering the complexity of the background, we set the value of $r$ to 2 for all three data sets. For LELRP-AD, it used the same segmentation regions as EaSLRP, and according to the complexity of the background distribution, the number of background endmembers $r$ was set to 2, 3, and 5 for the metal plates data set, the buoy data set, and the car data set, respectively. The cardinality $c$ was set to 0.02 for all three data sets.

For LRASR, the number of clusters K was set to 15. The selected pixels P were set to 20. We set the regularization parameters β and λ as equal to 0.1 for all three data sets.

For the EaSLRP method, the scale parameter $\gamma$ was set as 0.5, 0.6, and 0.5 for the metal plates data set, the buoys data set, and the car data set, respectively, as shown in Figures 6 and 7. The background endmembers and the cardinality were the same as for the LELRP-AD method.

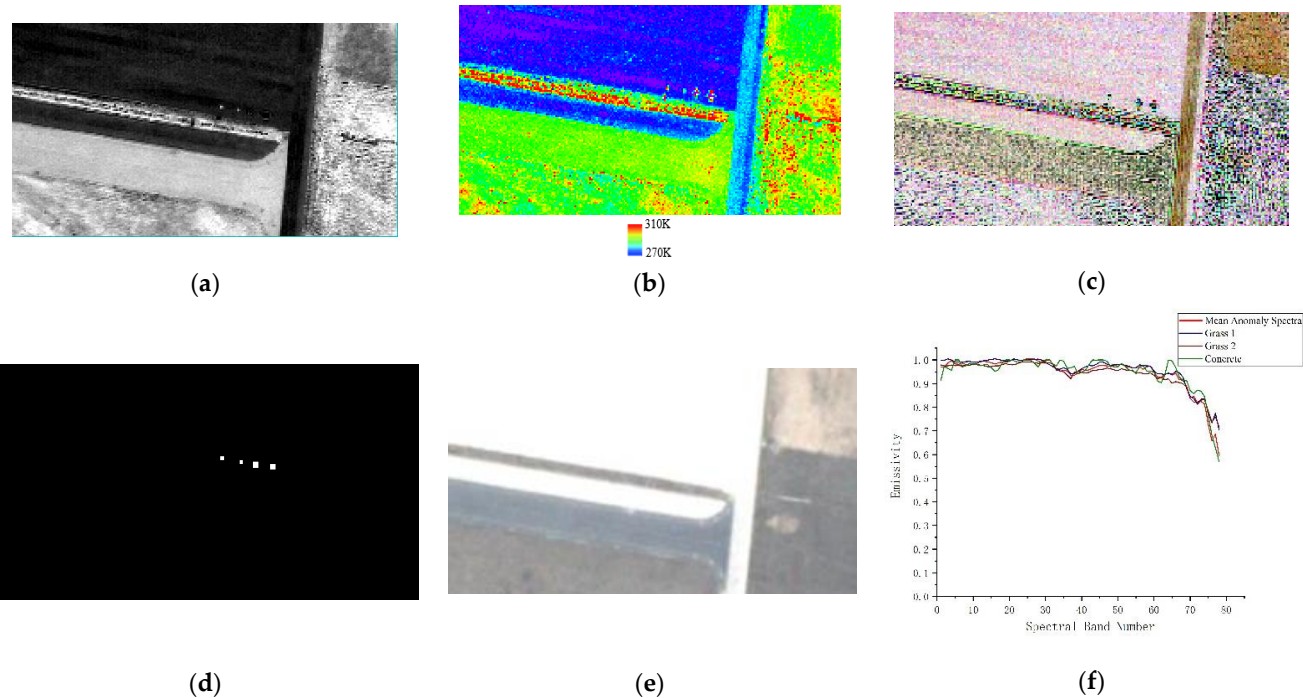

**Figure 6.** Metal plate data set. (**a**) Radiance domain; (**b**) temperature image; (**c**) emissivity RGB pseudo-color image synthesized by band 8, 11, 25; (**d**) ground-truth map; (**e**) RGB image; and (**f**) emissivity spectral signatures of anomalies and background.

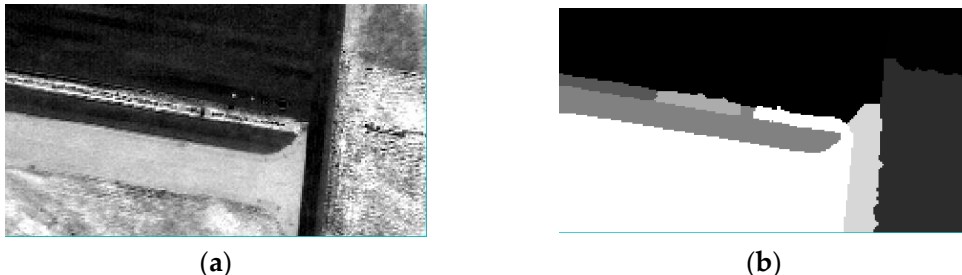

**Figure 7.** Metal plate data set. (**a**) Radiance domain image and (**b**) segmentation map.

### 3.3. Metal Plate Data Set and Detection Results

The LWIR hyperspectral image was collected near the runway of Zhengzhou Shangjie Airport and is shown in Figure 6. Four metal plates of man-made materials were defined as anomalies. The overall background distribution of the image was relatively simple.

MODTRAN was used to estimate the atmospheric upwelling radiance and atmospheric transmission of the radiance image obtained by the airborne sensor, and then the surface temperature and emissivity were retrieved by the FLAASH-IR TES method. The effectiveness of the emissivity image was verified by randomly selecting several points in the emissivity image and comparing their pixel vector with the corresponding material spectrum in the emissivity library. The radiance image and temperature image were used to generate homogeneous regions. The results are shown in Figure 7.

The results of the method testing are shown in Figure 8.

For the metal plates data set, it can be seen in Figure 8 that all the anomaly detection methods can detect the anomalies. The detection results illustrate that the background is suppressed well by the proposed EaSLRP method, and the effect of the anomaly and background separation is good. In the other results shown in Figure 8a–e, there is an obvious rectangular region that shows a strong response in the middle of the image. This region is a mixture of bare soil and grass, and it has a high emissivity and high spectral contrast with the surrounding environment. This is the reason why the other detectors

have difficulty in detecting this region as background. Compared to Figure 8a–e, it is clear from the result of the proposed method in Figure 8f that the proposed method can better suppress the complex background in the middle of the image.

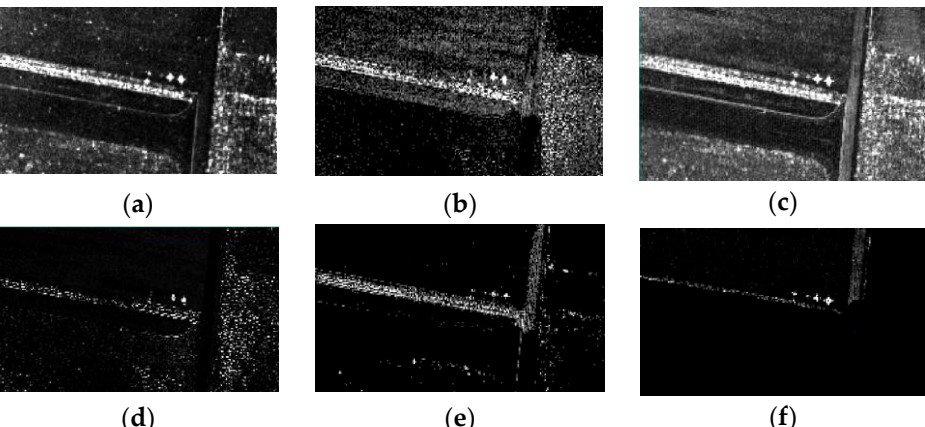

**Figure 8.** The target detection results of the different methods on the metal plates data set. (**a**) Reed—Xiaoli detector (RXD); (**b**) the segmented RX detector (SegRX); (**c**) the low-rank and sparse representation-based detector (LRASR); (**d**) the low-rank and sparse matrix decomposition (LRaSMD)-based Mahalanobis distance method (LSMAD); (**e**) the locally enhanced low-rank prior method (LELRP-AD); and (**f**) anomaly detection method combining emissivity and a segmented low-rank prior (EaSLRP).

### 3.4. Buoys Data Set and Detection Results

The LWIR hyperspectral image was collected around Dongguo Lake in Shangjie, Zhengzhou, Henan Province, China, and is shown in Figure 9. Two buoys on the lake are defined as anomalies. The background distribution of the image scene is more complex than the metal plates data set.

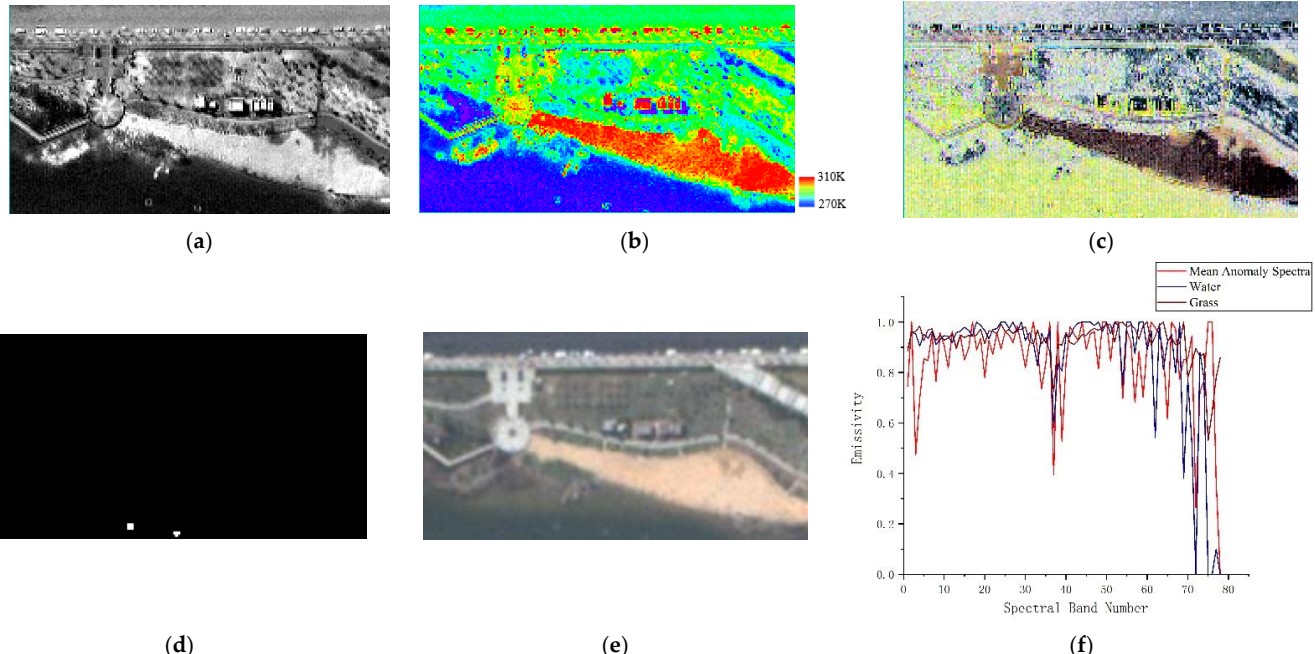

**Figure 9.** Buoys data set. (**a**) Radiance domain. (**b**) Temperature image; (**c**) emissivity RGB pseudo-color image synthesized by band 8, 11, 25; (**d**) ground-truth map; (**e**) RGB image; and (**f**) emissivity spectral signatures of anomalies and background.

After AC and TES, the radiance image and temperature image were used to generate homogeneous regions. Figure 10 shows the segmentation map.

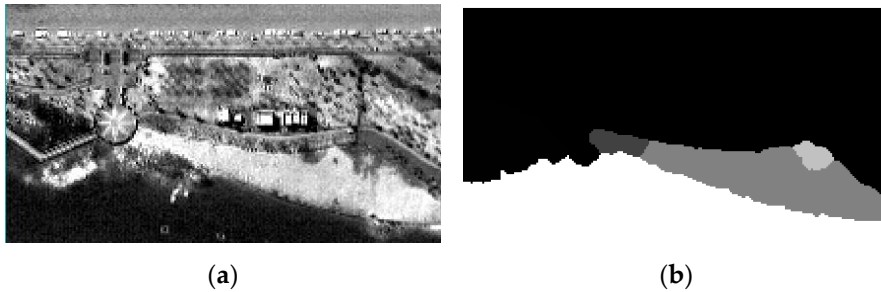

(**a**)                                                                                             (**b**)

**Figure 10.** Buoys data set. (**a**) Radiance domain image and (**b**) segmentation map.

The results of the method testing are shown in Figure 11.

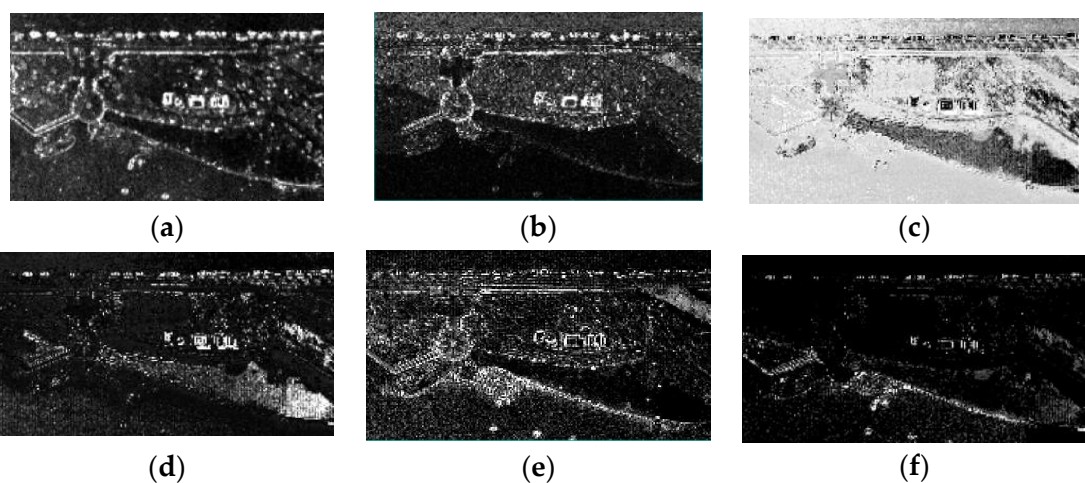

(**a**)                                           (**b**)                                           (**c**)

(**d**)                                           (**e**)                                           (**f**)

**Figure 11.** The target detection results of the different methods on the buoys data set. (**a**) RXD; (**b**) SegRX; (**c**) LRASR; (**d**) LSMAD; (**e**) LELRP-AD; and (**f**) EaSLRP.

For the buoys data set, it can be deduced in Figure 11 that SegRX, LSMAD, and LRASR fail to detect the anomalies, and a lot of background information is detected. The background of this image is complex, and if the whole image was used to calculate the background, many artificial targets that can be considered as noise would be detected. Therefore, it was necessary to divide the image into multiple homogeneous regions to estimate the background of each part accurately. Compared with the above methods, the proposed EaSLRP method can suppress the background well, even though the background distribution of the upper part of the emissivity image is very complex.

*3.5. Car Data Set and Detection Results*

The LWIR hyperspectral image was collected in an area next to a warehouse at Zhengzhou Shangjie Airport and is shown in Figure 12. The car on the grass is defined as an anomaly.

As in the previous experimental process, homogeneous regions were obtained according to the radiance image and temperature image. Figure 13 shows the result of the segmentation, where 12 parts are segmented.

For the car data set, the car has a high emissivity and low spectral contrast with the surrounding grass. Therefore, if the background estimation was not available, it would be difficult to distinguish the car from its surrounding environment. Therefore, it was necessary to segment the image to simplify the background and make the background information more accurate. The segmentation results increase the difference between the

target and the background, which is more conducive to detecting the target. According to the results shown in Figure 14, the SegRX method fails to detect the anomaly, and the contrast between the anomaly and background for the other methods is also low. Compared to these methods, the proposed EaSLRP method is better able to separate the background and anomaly.

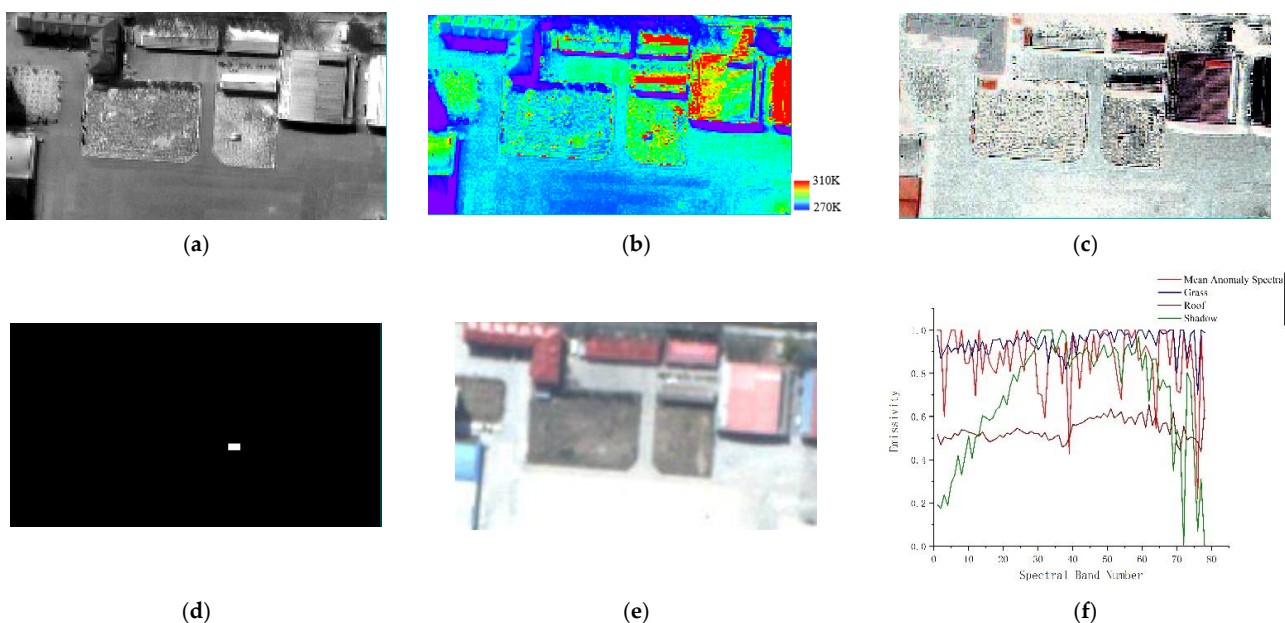

**Figure 12.** Car data set. (**a**) Radiance domain; (**b**) temperature image; (**c**) emissivity RGB pseudo-color image synthesized by band 8, 11, 25; (**d**) ground-truth map; (**e**) RGB image; and (**f**) emissivity spectral signatures of anomalies and background.

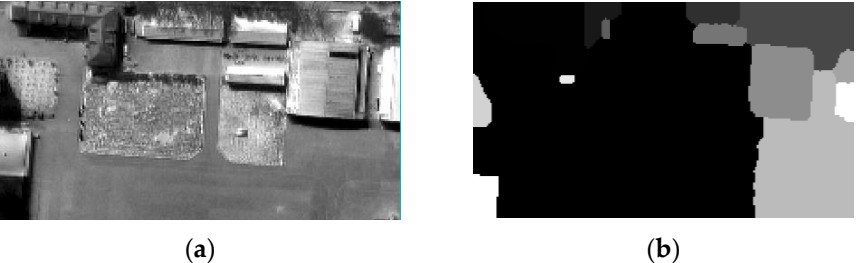

**Figure 13.** Car data set. (**a**) Radiance domain image and (**b**) segmentation map.

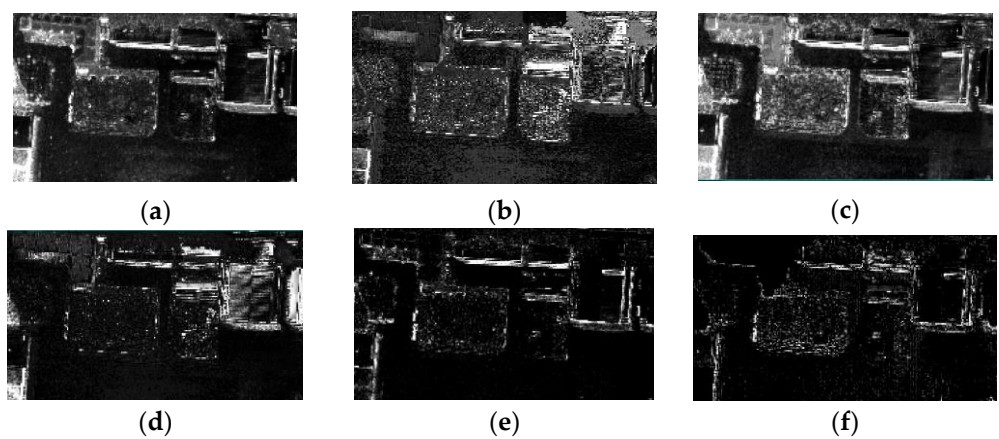

**Figure 14.** The target detection results of the different methods on the car data set. (**a**) RXD; (**b**) SegRX; (**c**) LRASR; (**d**) LSMAD; (**e**) LELRP-AD; and (**f**) EaSLRP.

### 3.6. Quantitative Evaluation

In order to assess the results quantitatively, receiver operating characteristic (ROC) [29] curves and the area under the ROC curve (AUC) [30] are employed. Figure 15 shows the ROC curves for these three experiments. The AUC values for the three data sets are listed in Table 1, where the bold numbers are the best results. Line charts of the AUC values are provided in Figure 16 for the different methods on all three data sets.

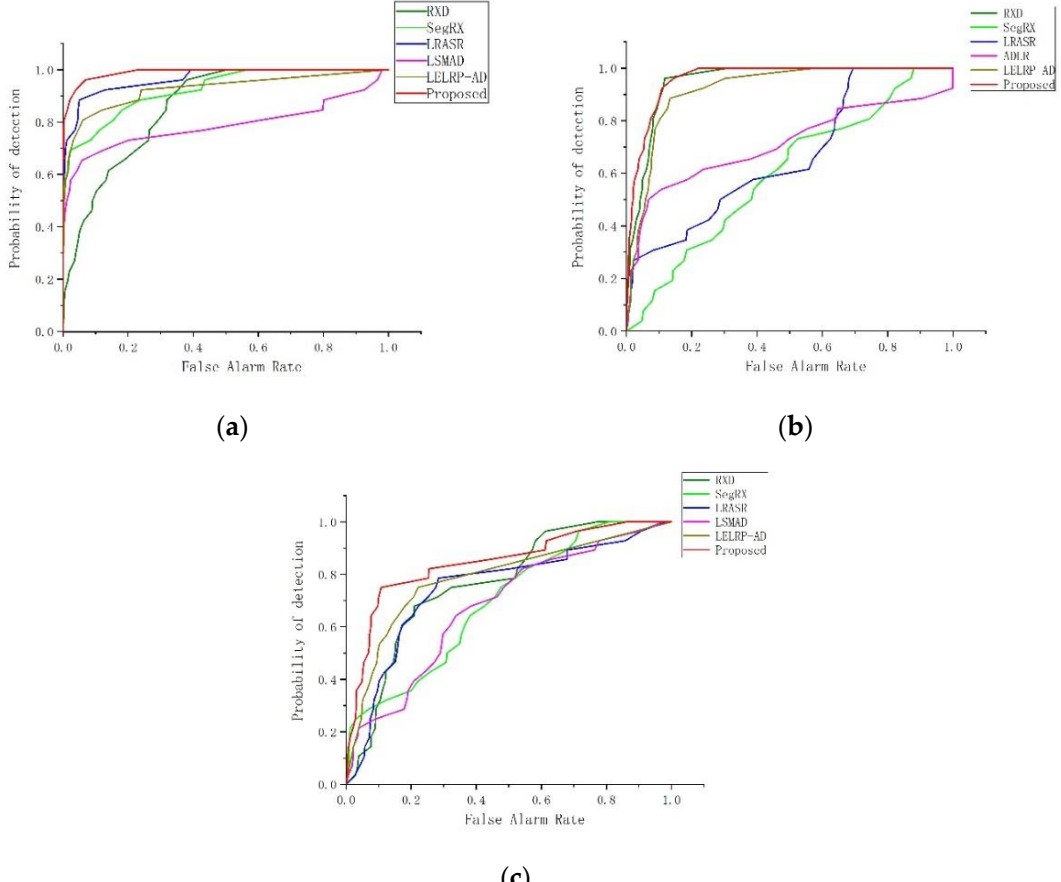

**Figure 15.** Receiver operating characteristic (ROC) curves. (**a**) ROC curves for the metal plates data set; (**b**) ROC curves for the buoys data set; and (**c**) ROC curves for the car data set.

**Table 1.** Area under the receiver operating characteristic curve (AUC) scores for the different methods on the three data sets.

| Data Sets | RXD [1] | SegRX [2] | LRASR [3] | LSMAD [4] | LELRP-AD [5] | Proposed |
|---|---|---|---|---|---|---|
| **Metal plates** | 0.8581 | 0.9240 | 0.9652 | 0.7879 | 0.9255 | **0.9896** |
| **Buoys** | 0.9493 | 0.6004 | 0.6634 | 0.5481 | 0.9200 | **0.9602** |
| **Car** | 0.7661 | 0.6934 | 0.7483 | 0.6786 | 0.7860 | **0.8457** |

[1] Reed–Xiaoli detector; [2] segmented RX detector; [3] sparse representation-based detector; [4] LRaSMD-based Mahalanobis distance method; [5] locally enhanced low-rank prior method.

For the metal plates scene, the ROC curve of the proposed method is always higher than that of the other methods. When the detection probability reaches 100%, EaSLRP's false alarm rate is 0.17, and EaSLRP's AUC value is 0.9896. Overall, the proposed method shows a competitive performance.

For the buoys data set, the proposed method's ROC curve is always higher than that of SegRX, LSMAD, LELRP-AD, and LRASR. Compared with the RXD method, the ROC curve of the RXD method is only higher when the false alarm rate is in the range of 0.06–0.14. When the detection probability reaches 100%, the RXD method's false alarm rate is 0.28,

and that of the proposed method is 0.20. The AUC value of the proposed method is 0.9602. In other words, the EaSLRP method presents the best detection effect.

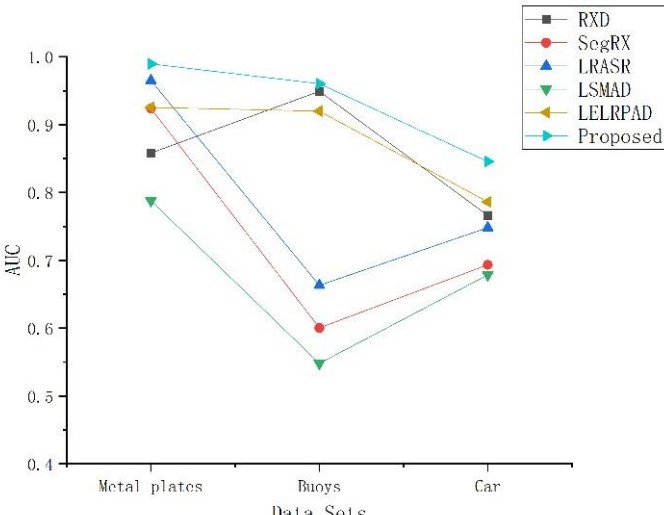

**Figure 16.** Line charts of the area under the receiver operating characteristic curve (AUC) scores.

For the car data set, the ROC curve of the proposed method is always above that of SegRX, LSMAD, LRASR, and LELRP-AD. Compared with the RXD method, EaSLRP's ROC curve is lower than that of the RXD method when the false alarm rate is between 0.52 and 0.85. When the detection probability reaches 100%, the false alarm rate of the RXD method is 0.72, and the false alarm rate of the proposed method is 0.85. The AUC scores of the RXD method and the proposed method are 0.7661 and 0.8457, respectively. That is to say, similar to previous experiments, the proposed method shows the best anomaly detection performance, followed by LELRP-AD.

Figure 15 shows the line charts of the AUC scores for the different detection methods. For all three data sets, the AUC score of EaSLRP is always the highest. The experimental results illustrate that the proposed EaSLRP method shows superior performance for thermal infrared hyperspectral image anomaly detection, especially for images with low spectral contrast or a complex background distribution. The results obtained for the car data set and the metal plate data set also demonstrate that the more complex the background distribution is, the more advantageous the proposed method is for anomaly detection.

The EaSLRP method also shows a superior background suppression effect in the scenario of a complex background. It simplifies the background by image segmentation and uses the low-rank prior method to remove some of the noise, which improves the purity of the final background estimation. The estimation background can increase the difference between the target and the background and makes the target more easily detected.

## 4. Discussion

### 4.1. Parameter Analysis for the Proposed Method

To verify the stability of the algorithm, it was necessary to analyze the main parameters of the algorithm, and the influence of the parameter change on the detection effect was observed. The main parameters of the EaSLRP method are the scale parameter $\gamma$ of the Potts-based method and the number of background endmembers $r$ in the local area. The metal plates and car data sets were selected for the sensitivity analysis. The AUC score is taken as the evaluation index.

The first analysis is of the scale parameter $\gamma$. According to experience, to cover the range from over-segmentation to under-segmentation, the value of $\gamma$ was set over the ranges of {0.02, 0.05, 0.1, 0.2} and {0.2, 0.3, 0.4, 0.5} for the metal plates data set and the car data set, respectively. Therefore, the emissivity images were divided into {19, 7, 5, 2} and {21, 15,13,10} local homogenous regions. As shown in Figure 17, when the background

endmembers *r* is constant, as long as the number of segmented homogeneous regions is kept in a normal range, the value of the scale parameter $\gamma$ hardly affects the experimental results. With the change of the scale parameter $\gamma$, the AUC score changes very little.

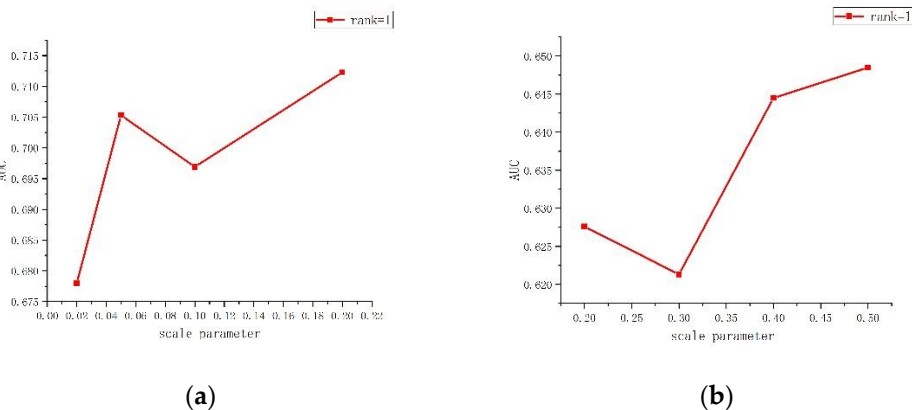

**Figure 17.** Analysis of the scale parameter $\rho$. (**a**) The metal plates data set and (**b**) the car data set.

For the number of background endmembers *r*, its value is generally not very large. This is because, after the original image is segmented into multiple homogeneous regions, the background composition of each local homogeneous region is relatively simple. According to experience, the value of *r* was varied over the range of [1,5]. Therefore, in the analysis, we increased the value of *r* from 1 to 5 in the process of anomaly detection for the metal plates and car data sets, and we observed the change of the AUC score, as shown in Figure 18. In Figure 18, it can be seen that, for the metal plates data set, when *r* is greater than 2, the AUC score does not change dramatically with the change of *r*. However, for the car data set, with the change of *r*, the AUC score changes greatly. In images with a simple background distribution, such as the metal plates data set, the number of background endmembers in homogeneous regions is less, so the AUC score changes only slightly with the increase of *r*. For an image with a complex background distribution, such as the car data set, the number of background endmembers in the different homogeneous regions can vary greatly, and the AUC score fluctuates greatly with the change of *r*. The value of *r* can greatly affect the experimental result, so it is necessary to try different values of *r* to find the most appropriate setting.

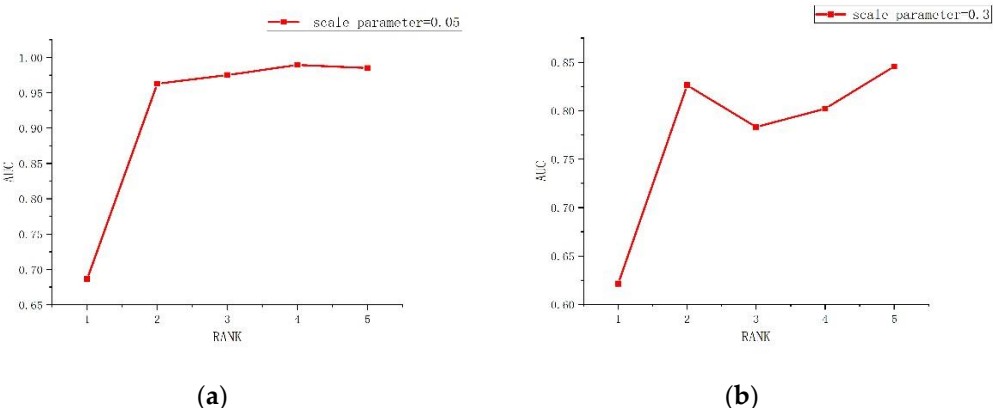

**Figure 18.** Analysis of the value of *r*. (**a**) The metal plates data set and (**b**) the car data set.

To sum up, the number of background endmembers *r* has the greatest impact on the detection effect of the EaSLRP method, while the number of homogeneous regions in the region segmentation has only a limited impact on the detection effect.

### 4.2. Radiation Domain and Emissivity Domain Analysis for the Proposed Method

To analyze the impact of data differences in the different domains on the detection results further, the radiance domain image and emissivity domain image were separately used to compare the detection results. The radiance domain image is the original image obtained by the sensor, and the emissivity domain image is the original image obtained after AC and TES. The spectral values of emissivity domain images are mostly in the range of 0.8–1.0, and the spectral contrast is generally lower than the pixel spectral values of radiance domain images. In order to analyze which image the proposed method can obtain a better anomaly detection result with, we analyzed and compared the performance with two groups of data from different images.

For the metal plates data set, as shown in Figures 19 and 20, the detection results in the emissivity domain are better in terms of the visual effect than those in the radiance domain. The ROC curve in the emissivity domain is always higher than that in the radiance domain. The AUC of the emissivity domain is 0.9896, and that of the radiance domain is 0.6380. This indicates that the detection effect of the emissivity domain is better than that of the radiance domain.

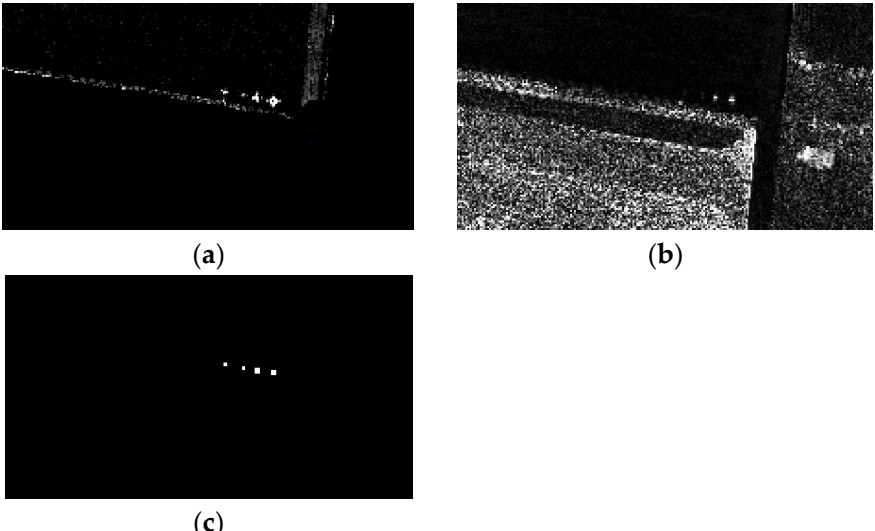

(**a**)

(**b**)

(**c**)

**Figure 19.** Metal plates data set. (**a**) Emissivity domain; (**b**) radiance domain; and (**c**) ground-truth map.

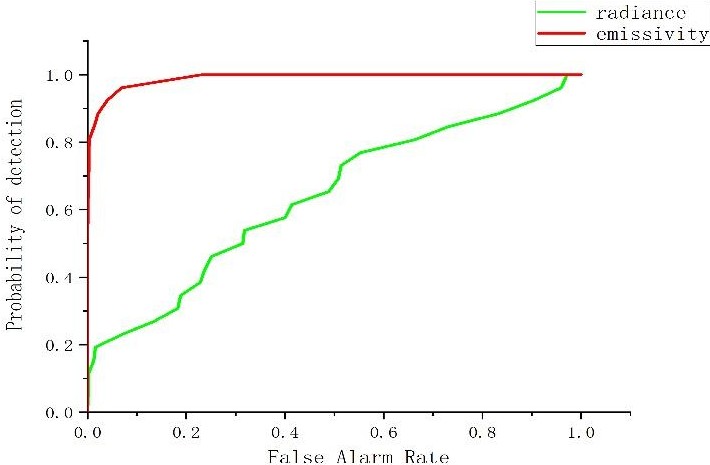

**Figure 20.** ROC curves of the metal plates data set.

For the car data set, as shown in Figures 21 and 22, when the false alarm rate is between 0.05 and 0.1, the ROC curve in the radiance domain is higher than that in the emissivity domain. The AUC value of the emissivity domain is higher than that of the radiance domain, and from the visual effect of the detection results, the emissivity domain detection image shows a better suppression effect on the central complex background area.

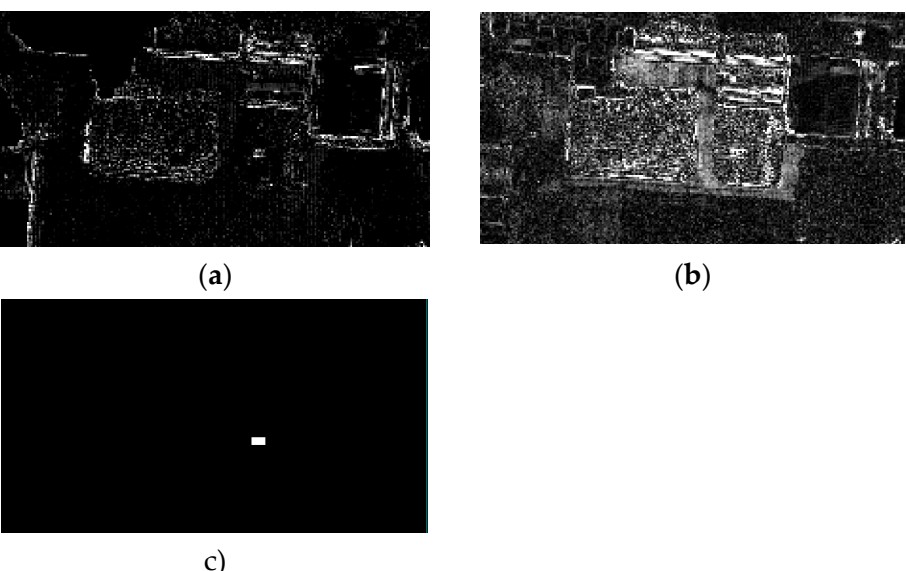

(a) (b)

c)

**Figure 21.** Car data set. (**a**) Emissivity domain; (**b**) radiance domain; and (**c**) ground-truth map.

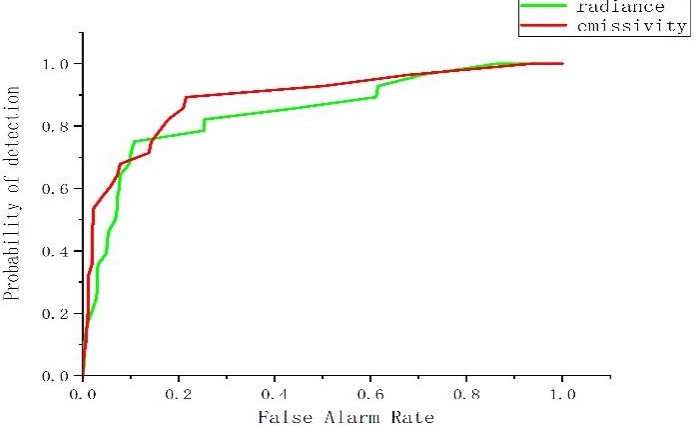

**Figure 22.** ROC curves of the metal plates data set.

In conclusion, it is more stable and effective to detect anomalies in the emissivity domain. The reason for this is that emissivity is the essential property of matter and is not affected by other factors. However, the radiance image is determined by the surface emissivity and surface temperature. The influence of temperature can lead to the same material having different spectra, while different materials may have similar spectra, which has a great impact on the anomaly detection results.

## 5. Conclusions

In this paper, based on the characteristics of LWIR hyperspectral images, we have presented a new LWIR hyperspectral anomaly detection method named EaSLRP. The EaSLRP method focuses on the separation of anomalies and background in LWIR hyperspectral images with low spectral contrast and SNR. It makes full use of the original data and background information to separate the anomalies from the background. The proposed

method uses the LWIR hyperspectral radiance image and temperature image to segment the emissivity image into multiple local regions and applies the range of homogeneous regions to the image of the emissivity domain, which is equivalent to dividing the emissivity image into several homogeneous regions. The background endmembers are then extracted from the local homogeneous regions of the emissivity domain image to construct an enhanced matrix, which is used to enhance the anomaly sparsity of the local homogeneous regions. The GoDec method is introduced to decompose the local enhancement matrix and obtain low-rank background information, and the Mahalanobis distance detector uses the background component and the original data to detect anomalies. In this study, three groups of LWIR hyperspectral images were used to verify the effectiveness and superiority of the proposed algorithm. The experimental results confirmed that the proposed EaSLRP method can accurately separate the anomalies and background in LWIR hyperspectral images. Compared with both statistical methods and the latest methods, the proposed EaSLRP method shows a superior anomaly detection performance and is also more robust.

Through the experiments, we came to the following conclusions: (1) The proposed method shows an excellent detection capability in thermal infrared hyperspectral emissivity imagery in which the spectral contrast and SNR are low; (2) the proposed method reduces the complexity of the background by region segmentation, and low-rank modeling and decomposition are utilized to remove some of the noise and extract pure background information. The spectral difference between the anomalies and background is increased by the use of the Mahalanobis distance detector, which results in the proposed method having a better detection capability than the other methods; (3) only the number of background endmembers has a significant influence on the accuracy of the algorithm; and (4) because there is no temperature effect, the emissivity domain image is more suitable for anomaly detection than the radiance image, and the effect is more stable.

In LWIR imagery, the spectral contrast is higher in the radiance image. Therefore, how to extend the anomaly detection algorithm from the emissivity domain to the radiance domain and solve the anomaly detection problem in the radiance domain in the case of temperature uncertainty will be an important research question in the future.

**Author Contributions:** Conceptualization, L.C. and Y.Z.; Methodology, X.Z.; Software, X.Z.; Validation, S.W. and L.C.; Formal analysis, S.W.; Investigation, X.Z.; Resources, L.C. and Y.Z.; Data curation, L.G.; Writing—original draft preparation, X.Z.; Writing—review and editing, L.C. and Y.Z.; Visualization, X.Z.; Supervision, L.C. and Y.Z.; Project administration, L.C. and Y.Z.; Funding acquisition, L.C. and Y.Z. All authors have read and agreed to the published version of the manuscript.

**Funding:** This research was funded by the National Key Research and Development Program of China under Grant 2017YFB0504202, in part by the National Natural Science Foundation of China under Grant 41771385 and 42071350, and by Postdoctoral Research Foundation of China.

**Data Availability Statement:** The data presented in this study are not available.

**Acknowledgments:** We thank anonymous reviewers for their insightful advice.

**Conflicts of Interest:** The authors declare no conflict of interest.

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
