# Peer review of "Anomaly Detection in Airborne Fourier Transform Thermal Infrared Spectrometer Images Based on Emissivity and a Segmented Low-Rank Prior"

_remotesensing, doi:10.3390/rs13040754_

Round 1

Reviewer 1 Report

In this manuscript, the authors proposed a new anomaly detection method combining emissivity and a segmented low-rank prior (EaSLRP) for use with longwave infrared (LWIR) hyperspectral imagery. The proposed EaSLRP method is divided into three parts: temperature/emissivity retrieval, extraction of the thermal infrared hyperspectral background information, and Mahalanobis distance detection. The evaluation of the proposed method was based on three groups of LWIR hyperspectral images. The experimental results showed that the proposed EaSLRP method can accurately separate the anomalies and background in LWIR hyperspectral images and that it outperforms other similar methods.

The manuscript is well written and the results are promising, so my advice is to be accepted for publishing in the journal after minor revision. For the revision, I would suggest that authors should introduce the application of hyperspectral anomaly detection in general and their method in particular. I think this is important for a wider audience of the journal to understand the scope and the importance of the presented research.

Author Response

Concern # 1: For the revision, I would suggest that authors should introduce the application of hyperspectral anomaly detection in general and their method in particular.

Response: Many thank you for your suggestion. The application of hyperspectral anomaly detection and our method advantage has been added in section I Introduction in in lines 41 to 44 and 48, which was highlighted in yellow.

Reviewer 2 Report

This reviewer would like to congratulate the authors for their work. Due to the complexity of the presented algorithm, involving a few substeps, further clarifications and modifications of the manuscript are required.

First I will focus on the style and the writing:

  • The shape of the document is quite strange, why is there that much of space on the left hand side?
  • Lines 127 and 133, please rewrite the expressions "In the part of ..."
  • In lines 151 to 153, the statement is not properly phrased. I would suggest to remove the line "The k-th band of the image is also composed of anomalies and background" as it does add more confusion than content. Otherwise, please rewrite.
  • Line 175, please reconsider the use of the word “However”. The fact that the background has low-rank characteristics is not against the fact that the proportion of anomaly pixels is small.
  • Line 183, please rephrase “when detecting in the global image”. Suggestion “when processing the global image with a/the target detection algorithm”.
  • Line 331, what does it mean “on a stabilized platform”? Do you mean on a “stable platform” or something else?
  • Line 334 is not clear. This reviewer assumes that the radiative transfer model is applied to the raw data captured by the camera. Please explain or rephrase.
  • Line 349, correct the word “chose”.

And some comments about content and structure of the manuscript:

  • The dataset images are blurry. Are there not some better pictures to show at least when they are introduced the first time? Maybe an RGB shot that was taken together with the LWIR acquisition.
  • The preprocessing that takes place into the data is not clear. This reviewer understands that the hyperspectral image has 81 bands (raw data). This is then converted to radiance thanks to the radiometric calibration of the sensor, but it still provides 81 bands, 78 after some of them are eliminated. What happened next? The emissivity and temperature are calculated, but those are just one value per pixel right?
  • Datasets are presented with 4 shots. Radiance domain should indicate the wavelength and/or band that is being displayed. The temperature and emissivity images may use a colormap display.
  • How was the Ground-truth map obtained? Is it somehow given? Please clarify.
  • The manuscript talks about hyperspectral imagery, but there is no single figure that let the user see any signature. When the datasets are being introduced an additional plot should be added with some key pixels’ signatures.

Author Response

Concern # 1: The shape of the document is quite strange, why is there that much of space on the left hand side?

Response: We prepared the manuscript in accordance with the latest format template of remote sensing 2021.

Concern # 2: Lines 127 and 133, please rewrite the expressions "In the part of ..."

Response: According to your suggestion, the expressions "In the part of ..." have been rewritten.

Concern # 3: In lines 151 to 153, the statement is not properly phrased. I would suggest to remove the line "The k-th band of the image is also composed of anomalies and background" as it does add more confusion than content. Otherwise, please rewrite.

Response: Thanks for your suggestion. The line "The k-th band of the image is also composed of anomalies and background" has been removed.

 Concern # 4: Line 175, please reconsider the use of the word “However”. The fact that the background has low-rank characteristics is not against the fact that the proportion of anomaly pixels is small.

Response: The word “However” has been replaced according to your suggestion.

 Concern # 5: Line 183, please rephrase “when detecting in the global image”. Suggestion “when processing the global image with a/the target detection algorithm”.

Response: Your suggestion has been integrated in our revision, thanks.

Concern # 6: Line 331, what does it mean “on a stabilized platform”? Do you mean on a “stable platform” or something else?

Response: The words “on a stabilized platform” have been replaced by “on a stable platform”.

Concern # 7: Line 334 is not clear. This reviewer assumes that the radiative transfer model is applied to the raw data captured by the camera. Please explain or rephrase.

Response: Thanks for your suggestion. The radiative transfer model is not applied to the raw data captured by the camera. We think that the sentence “The RTM was used for the imaging” is confused, and delete it.

Concern # 8: Line 349, correct the word “chose”.

Response: The word “chose” has been modified to “chosen”.

Concern # 9: The dataset images are blurry. Are there not some better pictures to show at least when they are introduced the first time? Maybe an RGB shot that was taken together with the LWIR acquisition.

Response: Your suggestion has been integrated in our revision, thanks. The RGB image of all datasets has been added. You can find in Fig6(e), Fig9(e), Fig12(e).

Concern # 10: The preprocessing that takes place into the data is not clear. This reviewer understands that the hyperspectral image has 81 bands (raw data). This is then converted to radiance thanks to the radiometric calibration of the sensor, but it still provides 81 bands, 78 after some of them are eliminated. What happened next? The emissivity and temperature are calculated, but those are just one value per pixel right?

Response: After removing the noisy bands, 78 bands were retained. Then temperature-emissivity separation algorithm was executed and a temperature image and a 78-channel emissivity image were obtained. The anomaly target emissivity spectrums are shown in Fig6(f), Fig9(f), Fig12(f).

Concern # 11: Datasets are presented with 4 shots. Radiance domain should indicate the wavelength and/or band that is being displayed. The temperature and emissivity images may use a colormap display.

Response: Thank you for your suggestion. The temperature images and emissivity images are displayed using color graphs, which are shown in Fig3(b), Fig6(b)~(c), Fig9(b)~(c), Fig12(b)~(c).

Concern # 12: How was the Ground-truth map obtained? Is it somehow given? Please clarify.

Response: The Ground-truth maps are based on field investigation and labeled with recorded longitude and latitude information combined with the aerial flight images. We have clarified in section 4.1 in line 349-351.

Concern # 13: The manuscript talks about hyperspectral imagery, but there is no single figure that let the user see any signature. When the datasets are being introduced an additional plot should be added with some key pixels’ signatures.

Response: Thanks for the suggestion and your suggestion has been integrated. Emissivity spectral signature images have been added and shown in Fig6(f), Fig9(f), Fig12(f).

Round 2

Reviewer 2 Report

Dear authors,

Thank you for having considered the suggestions made by this reviewer. Good work!

Kind Regards